# Dose-Dependent Transcriptional Response to Ionizing Radiation Is Orchestrated with DNA Repair within the Nuclear Space

**DOI:** 10.3390/ijms25020970

**Published:** 2024-01-12

**Authors:** Garima Chaturvedi, Avital Sarusi-Portuguez, Olga Loza, Ariel Shimoni-Sebag, Orly Yoron, Yaacov Richard Lawrence, Leor Zach, Ofir Hakim

**Affiliations:** 1The Mina and Everard Goodman Faculty of Life Sciences, Bar-Ilan University, Building 206, Ramat Gan 5290002, Israel; avital1005@gmail.com (A.S.-P.);; 2Institute of Oncology, Sheba Medical Center, Ramat Gan 5262000, Israel; ariel.sebag@gmail.com (A.S.-S.);; 3Institute of Oncology, Tel Aviv Soraski Medical Center, Tel Aviv 6423906, Israel

**Keywords:** GBM, DNA damage response, ionizing radiation, TADs, transcription, genome architecture

## Abstract

Radiation therapy is commonly used to treat glioblastoma multiforme (GBM) brain tumors. Ionizing radiation (IR) induces dose-specific variations in transcriptional programs, implicating that they are tightly regulated and critical components in the tumor response and survival. Yet, our understanding of the downstream molecular events triggered by effective vs. non-effective IR doses is limited. Herein, we report that variations in the genetic programs are positively and functionally correlated with the exposure to effective or non-effective IR doses. Genome architecture analysis revealed that gene regulation is spatially and temporally coordinated with DNA repair kinetics. The radiation-activated genes were pre-positioned in active sub-nuclear compartments and were upregulated following the DNA damage response, while the DNA repair activity shifted to the inactive heterochromatic spatial compartments. The IR dose affected the levels of DNA damage repair and transcription modulation, but not the order of the events, which was linked to their spatial nuclear positioning. Thus, the distinct coordinated temporal dynamics of DNA damage repair and transcription reprogramming in the active and inactive sub-nuclear compartments highlight the importance of high-order genome organization in synchronizing the molecular events following IR.

## 1. Introduction

Glioblastoma multiforme (GBM) is the most common primary malignant brain tumor in adults. The standard adjuvant treatment involves irradiation together with chemotherapy and some additional interventions [1]. However, these interventions and recent advances in cancer treatment have a marginal effect on median overall survival duration, which is currently 16 months [1,2]. The IR dose affects the magnitude and ratios of different DNA lesions as base damage, single-strand DNA breaks (SSBs) and double-strand DNA breaks (DSBs). DSBs, which are much harder to repair than other damage processes, are thought to be the most deleterious lesions induced by IR and a major factor leading to cell death. Irradiation-induced DNA damage leads to the activation of DNA damage response (DDR) pathways. This response is orchestrated by a network of proteins that coordinate cell cycle arrest and facilitate DNA repair processes to maintain genomic integrity [3]. One of the earliest events in DDR is the phosphorylation of histone H2AX proteins at the sites of DSBs, leading to the formation of γH2AX foci [4]. These foci reach their peak concentration within 30 min following irradiation [5]. Additionally, global DNA damage triggers the rapid and transient global repression of transcription, which is resolved within approximately two hours [6,7].

Since exposure to high doses of IR to treat GBM results in growth inhibition and cellular death, such treatments elicit time-dependent modifications in the transcriptional programs associated with cellular survival mechanisms, including p53, apoptosis, and nuclear factor kappa B (NF-κB) [8,9,10]. These pathways are also involved in the response of CAL51 breast cancer cells to IR [11], underscoring the critical role of transcriptional regulation in determining and reflecting the cellular fate and the effects of IR. However, the precise assessment of tumor volume and the incorporation of intrinsic factors of irradiation pose significant challenges to treatment [12,13]. Consequently, the cancer cells within the tumor and the surrounding tissue margins may be exposed to varying doses of IR, resulting in disparate cellular [14] and potentially transcriptional outcomes. These transcriptional variations can impact the treatment efficacy, potentially contribute to tumor relapse, and be used to identify biomarkers for radiation response; yet, they have been insufficiently explored.

The three-dimensional (3D) organization of the human genome plays a pivotal role in the proper regulation of transcription and DNA repair [15,16]. At the local organization scale, the segments of genomic loci interact more frequently within themselves than with the other regions, giving rise to topologically associating domains (TADs) that facilitate the association of genes with their regulatory elements. On a larger scale, TADs cluster in the active or inactive spatial sub-nuclear compartments [17]. While much attention has been given to the local scale [18,19,20], less is known about the dose-dependent effect of IR on the temporal and spatial coordination of DNA repair and transcriptional reprogramming occurring irrespective of the specific sites of DNA damage and after the majority of DNA has been repaired. 

In this study, we showed that following effective or non-effective IR doses, the genetic program evolves in a dose-dependent manner. Furthermore, the transcriptionally responsive gene loci are pre-positioned in active spatial nuclear compartments, irrespective of their transcriptional level or state. The radiation-dependent transcriptional reprogramming follows DNA damage repair in the active compartments, in conjunction with shifting the repair activity to the heterochromatic compartments, indicating spatial and temporal coordination between DNA repair and transcription. 

## 2. Results

### 2.1. Transcriptional Variations and Dynamics Depend on IR Dose

A single instance of exposure to a moderate dose of IR in the range of 1 Gy causes DNA damage that can be repaired, allowing approximately half of the cells within the population to resume their cell cycle and stay alive (Figure 1A). However, under higher doses in the 6 Gy range, the DSBs cannot be completely repaired or repaired aberrantly, leading to mitotic catastrophe and cell death by apoptosis, necrosis, or senescence [21,22,23]. To study the molecular events related to ineffective and effective IR treatments, U251 cells were exposed to 1 Gy or 6 Gy X-irradiation, respectively.

The irradiation of U251 cells resulted in a dose-dependent increase in γH2Ax foci 30 min post exposure, followed by a decline nearly to the basal levels at 24 h post IR (Figure 1), indicating the efficient repair of DNA DSBs. In order to investigate the transcriptional response to IR that extends beyond DNA repair processes and encompasses downstream cellular changes, RNA samples were collected at two specific time points. The first collection was performed 6 h following irradiation, corresponding to a stage where approximately 15–30% of the initial DSBs remained unrepaired [24,25,26]. The second collection was conducted at 24 h post IR.

One Gy IR had a marginal effect on gene expression within the first 6 h, but after 24 h, 183 genes were downregulated, and 459 were upregulated (LogFC > 0.5, *p* < 0.05) (Figure 2A). Treating the cells with a higher dose of 6Gy induced variation in the gene expression already within 6 h, which further increased over 24 h. Thus, a higher IR dose corresponds with a higher number of differentially regulated genes. K-mean cluster analysis identified four clusters of genes with similar transcriptional kinetics. Clusters 2 and 4 contained time-dependent differentially expressed genes, while clusters 1 and 3 had dose-dependent genes (Figure 2B). Pathway enrichment analysis using IPA [27] and ShinyGo [28] revealed that genes that were downregulated 24 h post 1 and 6Gy IR (cluster 2) were enriched for cancer-promoting pathways, including the TGF-β signaling pathway, which promotes cell growth and metastasis in the U251 cells [29]. The genes that were activated after 24 h of both the treatments (cluster 4) were enriched for cholesterol biosynthesis pathway in line with the response to a wide range of IR doses in multiple cancer types [30,31] (Figure 2C). The genes that were activated after 24 h, specifically in the higher-IR-dose treatment (cluster 3), showed similar pathways, suggesting that these pathways are enhanced in 6 Gy vs. 1 Gy IR. Notably, the genes that were downregulated already 6 h following 6 Gy IR (cluster 1) encoded genes that regulate the G2/M DNA damage checkpoint, the mitotic roles of Polo-like kinase (Plk), cyclins, and DNA damage-induced 14-3-3σ signaling. The downregulation of the cell cycle checkpoint is characteristic of cells undergoing apoptosis or death after cell division, as they enter the M phase before repairing their DNA [32]. The expression profiles are in line with the response to non-effective or effective IR doses in the 1 Gy- and 6 Gy-treated cells, respectively.

### 2.2. Subtle Changes in Genome Architecture after IR

The role of nuclear architecture is linked to transcription regulation and DNA repair. Hence, we investigated how genome organization responds to varying levels of irradiation both at high and low doses.

To capture the spatial environment of the IR-responsive genes, we used 4C-seq, which measures the genome-wide chromosomal associations with a point of interest (bait) [33]. As baits, we selected the *BTG2*, *PRSS35*, and *TRIM29* genes, which were activated 24 h following the two IR doses (cluster 4), along with *PPIB* and *HCRTR2*, which were stably expressed or silenced, respectively (Figure 3A). *BTG2* has roles in cell cycle control, cell differentiation, proliferation, DNA damage repair, and apoptosis in cancer cells, and recently, it has been reported as a radio-responsive gene [34]. *TRIM29* serves as a scaffold to assemble DNA damage repair proteins into chromatin in DDR [35]. *PPIB* encodes cyclophilin B (CypB), which is an essential survival signal in the GBM cells expressed in many cases of malignant glioma [36]. *PRSS35* is highly expressed in brain cancer [37,38].

To capture the chromosomal interactome of these genes, high-complexity 4C-seq libraries were sequenced at a high read depth. The 4C-seq libraries captured intra- and inter-chromosomal contacts, which were defined based on a *p* score (=−log10 *p*-value) assigned to every HindIII site centered in a 100 Kb running window. Since the intra-chromosomal and inter-chromosomal associating domains are characteristically different in intensity and size, the HindIII sites with the top 5% and 10% scores of the inter- and intra-chromosomal data, respectively, were retrieved (FDR < 0.04). The clusters of more than 15 consecutive positive HindIII sites were defined as a contact domain (Appendix A). Overall, we identified 109 contact domains in cis, encompassing 100 kb. Strikingly, despite global dose-dependent DSB induction and transcriptional reprogramming, the whole-genome contact profiles were remarkably similar 6 or 24 h after exposure to the two IR doses for all the bait loci (Figure 3B). The subtle variations in the 4C signal that were observed may reflect small inter-chromosomal dynamics or measurement noise (Appendix A).

### 2.3. Irradiation-Activated Genes Pre-Positioned within Active Nuclear Compartments

To determine the features of the spatial environments of the IR-responsive genes and how they are related to the transcriptional response, we first calculated their gene density (Figure 4A). The active and inactive chromatin spatially segregate into spatial compartments A and B, respectively [39].

As expected, the spatial environment of the active *PPIB* gene is gene-rich, while the inactive *HCRTR2* resides in a gene-poor spatial environment. Notably, the IR-activated genes *BTG2*, *PRSS35*, and *TRIM29* predominantly reside in gene-rich spatial environments regardless of their expression level at resting cells. For example, *TRIM29* resides in a gene-rich environment, although before IR, it was expressed as low as *HCRTR2*, while *PRSS35* was expressed at a level that is ten folds higher (Figure 3A). In line with the previous studies [40], this suggests that gene regulation is a stronger determinant than expression for gene positioning in the A compartment. To further examine genome organization in terms of gene regulation, we calculated the relative enrichment of dynamically expressed genes in the spatial environments of the bait genes (Figure 4B). The spatial environment of the inactive gene *HCRTR2* showed the highest enrichment for genes from clusters 1 and 2, which were repressed following IR. In contrast, the environment of the active gene *PPIB* was primarily enriched with genes that were activated 24 h after 1 Gy and 6 Gy or 6 h following 6Gy IR (clusters 4 and 3, respectively). Notably, while the environments of the activated genes *BTG2*, *PRSS35*, and *TRIM29* were also depleted for IR-repressed genes, they showed an exceptionally high level of enrichment for IR-activated genes in their spatial environments (Figure 4B).

### 2.4. Transcriptional Response to IR Is Spatially and Temporally Coordinated with DNA Repair

While IR generates global DNA damage in both the transcriptionally active and inactive regions in the genome [41,42], several studies have indicated that the repair kinetics of heterochromatin are slower than those in the euchromatic regions [43]. To understand how DNA repair and gene expression dynamics in response to IR are coordinated within the nuclear space, we combined our spatial 4C-seq measurements of the U251 cells with the γH2AX ChIP-seq profiles following IR in the HeLa cells [20]. The levels of expression of our bait genes were overall similar in the U251 and HeLa cells, excluding *PRSS35*, which was active in U251, but silent in the HeLa cells (Appendix A). In addition, the chromatin decorations of the bait genes in the Hela cells correspond to the A and B compartments, suggesting that the features of the gene’s spatial environments are similar in both the cell lines (Appendix A) [44].

In the resting cells, the γH2AX signal level in the spatial environments of the irradiation-responsive *BTG2* and *TRIM29* and the active *PPIB* genes was higher than those in the environments of the silent genes *PRSS35* and *HCRTR2*. Three hours following IR, the γH2AX signal level was elevated predominantly in the compartments of the active and activated genes prior to their activation. Then, 24 h post IR, the γH2AX signal was depleted from the active compartment concomitantly with gene activation. We noted that the γH2AX signal level in the subnuclear compartment of the highly and constitutively active *PPIB* gene was higher than that in the compartments of the activated genes, but further analysis is required to determine whether this difference is functional.

The inactive B compartment can be divided into sub-compartments with distinct facultative and constitutive heterochromatin signatures. Constitutive H3K9me3-enriched heterochromatin forms at the pericentromeric regions and telomeres and remains stable across the cell types, while facultative heterochromatin correlates positively with H3K27me3 and negatively with H3K36me3 and is more dynamic and can lose its condensed state and become transcriptionally active in specific developmental or cellular states [39,45]. We found that the γH2AX signal level at the facultative heterochromatic sub-compartment of *HCRTR2* was lower than that in the active compartments in the resting cells and remained low 3 h post IR, but elevated in the active compartments. Twenty-four h post IR, while the genes were activated together with a reduction in γH2AX in the active compartment, the γH2AX signal level was increased in the *HCRTR2* sub-compartment, suggesting slower repair kinetics in response to IR in the facultative heterochromatin spatial environments (Figure 5), similarly to the constitutive heterochromatin regions [20] and similarly to both the heterochromatic sub-compartments in response to ultraviolet irradiation [46].

## 3. Discussion

Radiotherapy has a central role in the treatment of malignant brain tumors. Despite the advancements in tumor volume evaluation, treatment planning systems, and IR techniques, understanding the consequences of the exposure of tumor cells to varying IR doses remains a challenge [13]. In this study, we demonstrated the fundamentally distinguishable effect of effective and non-effective radiation doses on the genetic program of GBM U251 cells. The identification of changes in the population-averaged genomic data implies their consistent occurrence or presence in a substantial subset of cells within the population. This strongly suggests that the observed transcriptional changes reflect consistent and robust features of the genome’s response to irradiation, which are likely to be under biological control, despite the random distribution of DNA damage sites.

While both the non-effective 1Gy and effective 6Gy doses resulted in minor transcriptional changes 6 h after IR, these changes were enhanced after 24 h, particularly in the 6Gy dose. The progressive development of dose-dependent transcription programs following the repair of IR-induced DNA damage underscores the critical role of transcriptional regulation in determining the cellular fate and the effects of IR. The genes undergoing activation and repression in both the treatments after 24 h are associated with pathways characteristics of a cellular response to IR. The lower-dose-exposed cells are less likely to undergo apoptosis, leading to an increase in mutation burden, which may contribute to a future cancer relapse and progression. However, higher doses rapidly and specifically repressed the cell cycle regulatory pathways, promoting apoptosis. The genes that were activated in both the treatments after 24 h were associated with cholesterol biosynthesis [31], which has been linked to cytotoxicity. Additionally, polo-like kinase in mitotic processes (including the *CDC25C*, *PLK1*, *CCNB2*, *CDC20*, *PTTG1, CCNB1*, *KIF23*, and *PRC1* genes) is one of the primary pathways experiencing downregulation, suggesting the inhibition of cell division.

The sensitivity to radiation varies depending on the cancer type. However, the previous studies have reported that changes in gene expression in response to effective doses of IR show minimal dependence on the tissue of origin [47]. The radiation response appears to be influenced more by the integrity of the p53 and DNA repair pathways [9,11]. Therefore, we suggest that the dose-dependent and time-dependent transcriptional changes that we measured represent a fundamental response to IR. We found that nuclear architecture at the compartment level was stable 6 h and 24 h post 1Gy or 6Gy IR. Similar results were reported for fibroblasts and lymphoblastoid cells treated with 5Gy IR [18], indicating that high-order genome architecture at the compartment level is a stable component of genome function following IR.

By integrating the nuclear architecture with transcription profiles, we discovered a predisposition of IR-responsive genes towards specific subnuclear environments that align with their subsequent expression patterns. Specifically, the IR-activated genes exhibited a preference for active sub-nuclear compartments, which were also enriched with the other IR-activated genes. Conversely, the IR-repressed genes were found in the transcriptionally silent nuclear compartments. These findings suggest a potential functional relationship between the subnuclear localization of genes and their transcriptional response to IR. This organization differs from the spatial arrangements observed for the genes responsive to signaling hormones, such as the glucocorticoid, estrogen, and androgen receptors, where both the activated and repressed genes tend to be localized within active nuclear compartments [48,49]. Given that the transcriptional response to IR has cell-type-invariant features, it would be interesting to test whether the spatial organization of IR-responsive genes remains similar despite the cell-type-specific features of the genome architecture.

In summary, the kinetics of changes in the genetic program upon the IR of tumor cells is time- and dose-dependent. There are distinct coordinated temporal dynamics of DNA damage repair and transcription reprogramming in the active and inactive sub-nuclear compartments. The magnitude of IR dosage exerts an influence on the degrees of DNA damage repair and transcription modulation. Our research provides valuable insights into the dynamics of genetic reprogramming in the active and inactive subnuclear compartments in the face of high- and low-level genotoxic stress. The inter-relationship between DNA architectural changes with gene transcription changes may introduce a new direction for the development of radio-protective and radio-sensitization agents.

## 4. Materials and Methods

### 4.1. Cells and IR Treatment

U251 cells were cultured in Dulbecco’s modified Eagle’s medium (DMEM; Sigma-Aldrich Inc., St. Louis, MO, USA) supplemented with 10% FBS, 3.2% nonessential amino acids (NEEA), 4 mmol/L l-Glutamine, 100 U/mL penicillin, and 100 μg/mL streptomycin on six 14 cm plates. A total of 20 × 10^6^ cells were exposed to IR doses of 1 Gy and 6 Gy, while 18.5 × 10^6^ cells were maintained as the control group. The cells were harvested at 6 or 24 h post irradiation. The untreated cells were harvested 24 h after the irradiation treatment of the rest of the cells.

### 4.2. Clonogenic Assay

U251 cells were seeded in 6-well plates at specified densities, as outlined in Appendix A, with triplicates. The following day, the cells underwent radiation exposure at designated doses (0, 0.5, 1, 4, and 6 Gy). Following a 9-day incubation period, the colonies were treated with 0.5% crystal violet and counted using a microscope.

### 4.3. γH2AX Staining

U251 cells were grown in poly-lysine-coated coverslips in 6-well plates and either untreated or irradiated with 1 Gy or 6 Gy of gamma irradiation. The cells were harvested 30 min and 24 h post irradiation. Then, the slides were stained with gH2AX antibody and DAPI. High-resolution confocal images were taken and analyzed using LAS X image analysis software (https://www.leica-microsystems.com/ accessed on 19 June 2017)

### 4.4. RNA Extraction and RNA-seq

Total RNA was isolated from cells using an RNA purification kit (GeneAll) as directed by the manufacturer. The quality of the extracted RNA was assessed using a Bioanalyzer (Agilent, Santa Clara, CA, USA), and only the RNA samples with an RIN score exceeding 9 were utilized for library preparation. Messenger RNA (mRNA) was enriched from 1 μg of total RNA by Poly(A) mRNA Magnetic Isolation Module (New England Biolabs, Ipswich, MA, USA) following the manufacturer’s protocol. The NEBNext Ultra RNA Library Prep Kit (New England Biolabs) was used to create cDNA libraries in accordance with the manufacturer’s instructions. The concentration of the libraries was measured with a Qubit fluorometer (Invitrogen, Waltham, MA, USA) using the DNA High Sensitivity Kit (Invitrogen). The library quality and fragment sizes were evaluated using a Bioanalyzer or Tape station (Agilent). RNA-Seq libraries from a minimum of two biological replicates for each experimental condition were sequenced on the Illumina Hi-seq 2000 platform.

### 4.5. 4C-seq

The 4C (chromosome conformation capture) technique was carried out following previously described methods [50,51]. The cells were fixed using 2% formaldehyde for a duration of 10 min. The cross-linked chromatin was subjected to overnight digestion with an excess amount of HindIII enzyme (New England Biolabs). Subsequently, the DNA ends were ligated under diluted conditions that promote the formation of junctions between cross-linked DNA fragments. The ligated junctions were then circularized through digestion with DpnII (Thermo Scientific, Waltham, MA, USA). Chromosomal interactions with the target regions, known as baits, were amplified using inverse PCR primers (Appendix A) and Platinum Taq DNA Polymerase (LifeTechnologies, Carlsbad, CA, USA). The Illumina Hi-seq 2000 platform was used to sequence the 4C-seq libraries.

### 4.6. RNA-seq Analysis

The alignment was performed using TopHat [52]. The read count on transcripts was acquired using HTSeq [53]. Expression and differential analyses were performed using “edgeR” [54], which assigns to each gene a false discovery rate (FDR) value and calculates the log fold change (logFC) between the two conditions. Genes with FDR < 0.05 and logFC greater than |0.5| were considered differentially expressed. Functional enrichment was performed using IPA (Qiagen, Hilden, Germany [27]).

### 4.7. 4C-seq Analysis

The reads were sorted according to their barcodes to different fastq files for each bait and condition and aligned to the human (hg19) genome using BOWTIE [55]. The reads were then counted for each HindIII site. For domain detection, the number of reads on each HindIII site was counted in sliding windows of 50 Kb with 25 Kb steps. Hi-C heatmaps are from the 3D genome browser [56].

## 5. Conclusions

In summary, the kinetics of changes in the genetic program upon the IR of tumor cells is time- and dose-dependent. There are distinct coordinated temporal dynamics of DNA damage repair and transcription reprogramming in the active and inactive sub-nuclear compartments. The magnitude of IR dosage exerts an influence on the degrees of DNA damage repair and transcription modulation. Our research provides valuable insights into the dynamics of genetic reprogramming in the active and inactive subnuclear compartments in the face of high- and low-level genotoxic stress. The inter-relationship between DNA architectural changes with gene transcription changes may introduce a new direction for the development of radio-protective and radio-sensitization agents.

## Figures and Tables

**Figure 1 ijms-25-00970-f001:**
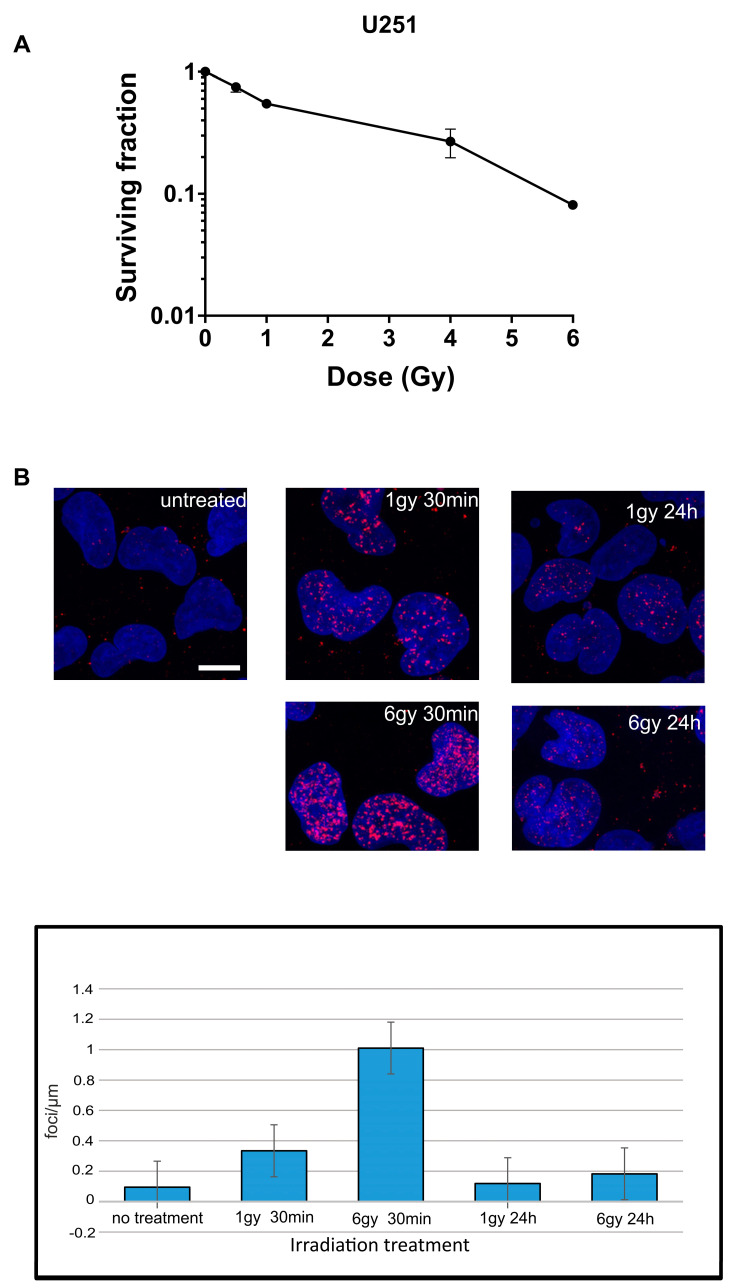
Clonogenic assay and U251 gH2AX cell staining for gH2AX. (**A**) Survival curve for U251 cells. (**B**) The upper picture shows stained U251 cells stained with gH2AX antibody at DAPI in untreated condition and at 30 min and 24h after exposure to 1 Gy or 6 Gy of gamma irradiation. Lower panel shows foci counts/μm^3^. The scale bar:10 μm.

**Figure 2 ijms-25-00970-f002:**
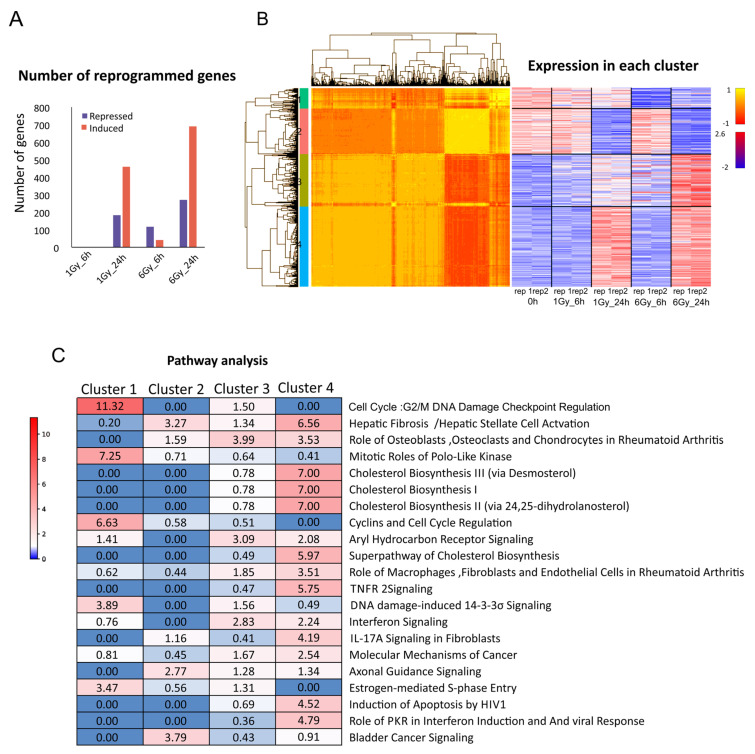
RNA-seq of irradiated U251 cells. (**A**) Number of genes that are reprogrammed after radiation in each time point and intensity relative to 0 h. Blue-repressed genes; red-induced genes. (**B**) Expression profile cluster analysis. Left-heatmap showing the correlation of the expression between each gene and all other genes, clustered into 4 major groups. Right-heatmap showing the expression (normalized and scaled cpm) of each gene in the 4 clusters. (**C**) Pathways analysis using IPA software (https://digitalinsights.qiagen.com/, accessed on 11 November 2018). Enrichment score is shown for each pathway in each cluster.

**Figure 3 ijms-25-00970-f003:**
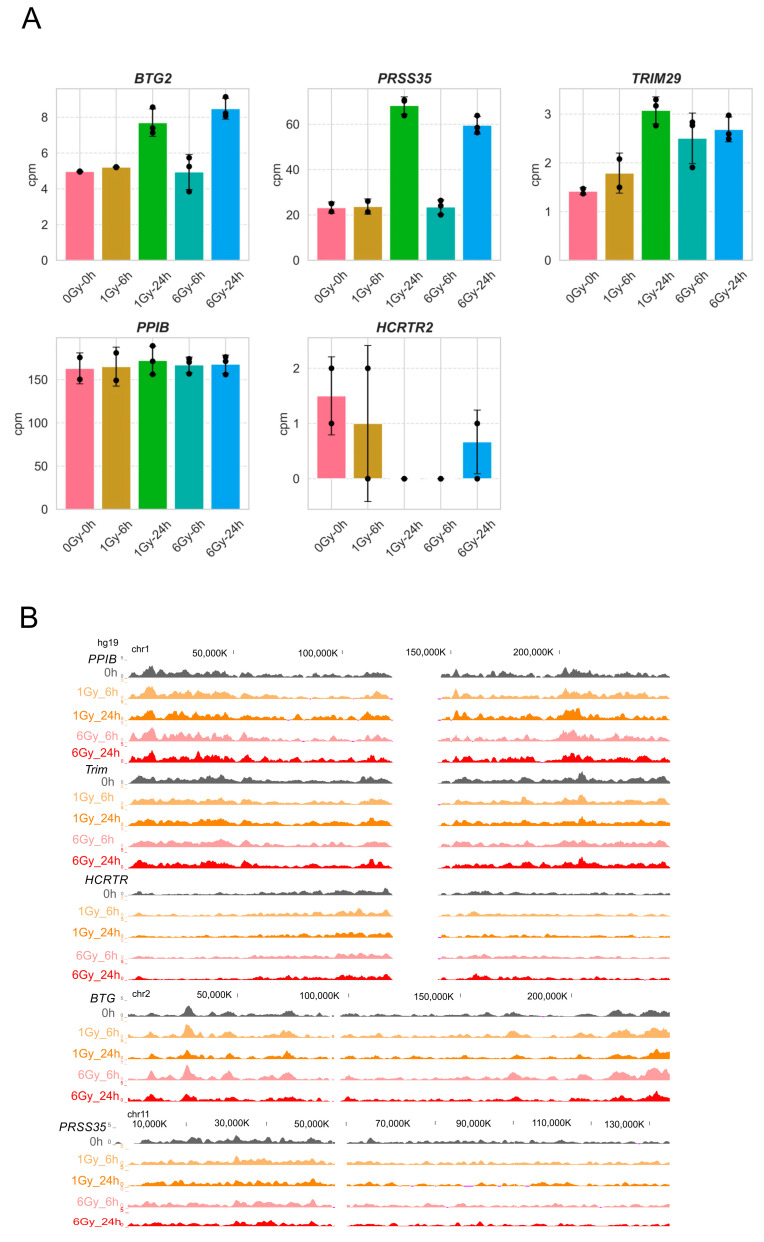
The 4C-seq of U521 cells. (**A**) Expression of the 4C-bait genes in resting and irradiated U251 cells. RNA was collected 6 and 24 h following 1Gy or 6Gy IR. Mean and standard deviation of CPM from three independent RNA-seq experiments are shown. (**B**) Spatial contacts of the indicated genes in resting and irradiated cells (genome build hg19).

**Figure 4 ijms-25-00970-f004:**
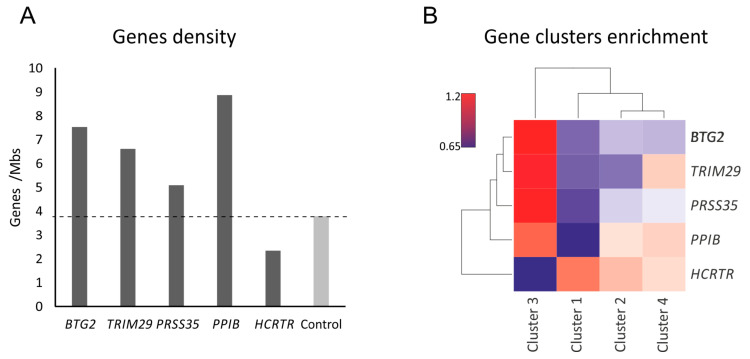
4C-seq and gene expression in U251 cells. (**A**) Genes density (calculated by genes per Mbs) in the spatial contacts of each bait and in the genome (control, light gray). (**B**) Heatmap showing the relative enrichment of genes from each cluster of expression in the spatial contacts of each bait.

**Figure 5 ijms-25-00970-f005:**
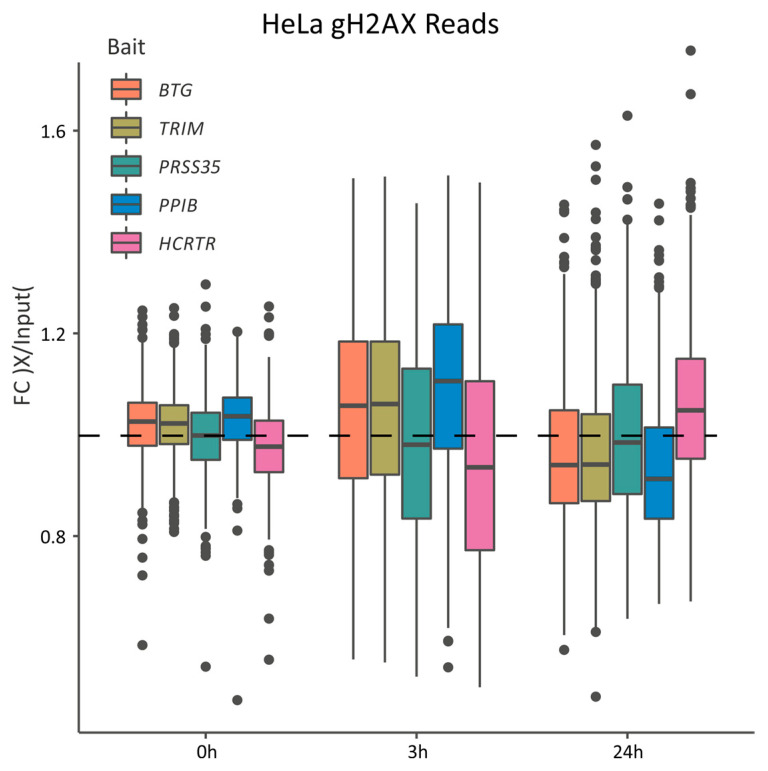
The 4C and γH2AX ChIP-seq data. The panel shows the signal of γH2AX ChIP (the number of reads from IP samples divided by the number of reads from the input sample) in the spatial contact loci of each bait.

## Data Availability

Data are currently being uploaded to NCBI. We are trying our best to provide accession number as quickly as possible.

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
