# Peer review of "Dose-Dependent Transcriptional Response to Ionizing Radiation Is Orchestrated with DNA Repair within the Nuclear Space"

_ijms, 2024, doi:10.3390/ijms25020970_

Round 1

Reviewer 1 Report

Comments and Suggestions for Authors

“Dose dependent transcriptional response to ionizing radiation 2 is orchestrated with DNA repair within the nuclear space” by Garima Chaturvedi and colleagues provides an overview on temporal dynamics of DNA damage repair and transcription reprogramming events following IR, highlighting the importance of the active and inactive sub-nuclear compartments localization. The use of 4C (chromosome conformation capture) technique is quite interesting as it gives indications on chromosomal interactions.

That said, the quality of this research is, in this reviewer opinion, sub-optimal and although the topic is interesting and worth exploring, substantial modifications and integrations are needed to make the paper worth of publication.

SPECIFIC COMMENTS:

1)      The authors stated that the process is orchestrated within the nuclear space by investigating the effect of IR on one cell line, with some additional data using HeLa cells. To give strength to the paper and to support their conclusions, the authors should validate their data by using other cell lines, ideally 2 other brain tumour cell lines.

2)      Sub-lethal and lethal IR doses need objective confirmation, ideally with clonogenic assay and viability assay. As GBM are generally radioresistant, 6Gy might not reflect a true lethal dose, also considering that 24h post 6Gy irradiation no gH2AX foci are detected in the cells.

3)      gH2AX foci data need some sort of validation, flow cytometry could be a viable option. Moreover, statistical analysis is needed.

4)      Figure 2B is difficult to read.

5)       In Results 2.2, it is unclear whether the analysis has been performed on GBM or HeLa cells. Since the basal expression of the reported genes is shown in HeLa in the Supplemental Figure 2, one could assume these are the cells used, and therefore why the GBM cells were not used?

6)      Figure 3A data need error bars in them to account for variation within experiments. Also, statistical analysis is needed.

7)      Figure 3B is illegible, and therefore make it impossible to draw conclusions.

8)      Figure 3A data need error bars in them to account for variation within experiments. Also, statistical analysis is needed.

9)      Once again, HeLa cells were used for gH2AX reads in Figure 5, why not the GBM?

Author Response

We thank you for taking the time to consider our manuscript and for the opportunity to revise it for publication. We would also like to thank the reviewers for their thoughtful feedback and suggestions. We have revised our manuscript according to the reviewers’ comments, and now present a revision that we feel is suitable for publication in IJMS.

Please see attached file for full detail.

We replicate the reviewer’s comments (black text) and provide our point-by-point responses inline (green text):

Reviewer 2 Report

Comments and Suggestions for Authors

Chaturvedi, et al. present a unique report on the spatio-temporal regulation of DNA damage following ionizing radiation dose delivery.  This study is unique in its design and the data being presented, however, there are limitations in its utility resulting from a lack of biological characterization within the model system.  The following should be addressed prior to publication: 

1. The author's should address the dose-dependency of single-stranded versus double-stranded DNA damage.   Does the spatial regulation depend on the type of DNA damage and pathway that is activated (ATM versus ATR)? 

2.  The author's use gamma-H2AX foci as a marker of DNA damage, however, ionizing radiation more prevalently induces single-stranded DNA damage rather than double-stranded damage.  Have you performed any validation analysis of the DNA damage that is being (i.e., neutral versus alkaline comet assays)? 

3. Using the two doses of IR described, how toxic were the doses (i.e. what was the clonogenic survival following dose delivery)? 

4. Given the enrichment of cancer-promoting pathways at 24 h post-IR, were there any changes in cell growth at this time point? Was there an activation of the G2/M checkpoint as suggested by the data?  

Comments on the Quality of English Language

No significant issues detected.

Author Response

(The authors gave the same response as above.)

Reviewer 3 Report

Comments and Suggestions for Authors

I consider this study to be very unique and novel.

Radiation-induced DNA damage is a fundamental principle of radiotherapy-induced cell death, and although various mechanisms for local repair of DNA damage have been described, knowledge of how gene expression is altered in the cell as a whole is limited. In this study, the authors examine temporal changes in RNA transcriptional regulation by lethal and sublethal radiation injury and show that the changes can be categorized by dose and time (regardless of the site of DNA damage).

We also have very limited knowledge of spatial changes in genomic structure (gene location in the nucleus). In this study, the authors examined changes in nuclear structure and found that the relative positions of gene loci were similar regardless of dose. Combined with the results of transcriptional responses, the authors infer a functional relationship between the nuclear localization of genes and the transcriptional response to irradiation.

I have one request.

Figures 2 and 3 are small and hard to see, can you make them larger or excerpt the important parts?

Author Response

(The authors gave the same response as above.)
